# Efficacy of an Intervention to Reduce Stigma Beliefs and Attitudes among Primary Care and Mental Health Professionals: Two Cluster Randomised-Controlled Trials

**DOI:** 10.3390/ijerph18031214

**Published:** 2021-01-29

**Authors:** Francisco José Eiroa-Orosa, María Lomascolo, Anaïs Tosas-Fernández

**Affiliations:** 1Section of Personality, Assessment and Psychological Treatment, Department of Clinical Psychology and Psychobiology, Faculty of Psychology, University of Barcelona, 08035 Barcelona, Spain; 2First-Person Research Group, Veus, Catalan Federation of 1st Person Mental Health Organisations, 08035 Barcelona, Spain; 3Obertament, Catalan Alliance against Stigma and Discrimination in Mental Health, 08010 Barcelona, Spain; mlomascolo@obertament.org (M.L.); anais.fernandez.tosas@gmail.com (A.T.-F.)

**Keywords:** activism, discrimination, mental health, primary care, participation, stigma

## Abstract

Although it may seem paradoxical, primary care and mental health professionals develop prejudices and discriminatory attitudes towards people with mental health problems in a very similar way to the rest of the population. The main objective of this project was to design, implement and evaluate two awareness-raising interventions respectively tailored to reduce stigmatising beliefs and attitudes towards persons with a mental health diagnosis among primary care (PC) and mental health (MH) professionals. These interventions were developed by Obertament, the Catalan alliance against stigma and discrimination in mental health. Activists from this organisation with lived experience of mental health diagnosis carried out awareness-raising interventions in PC and MH health centres. The Targeted, Local, Credible, Continuous Contact (TLC3) methodology was adapted to the Catalan healthcare context. The efficacy of these interventions was evaluated using two prospective double-blind cluster-randomised-controlled trials. Stigmatizing beliefs and behaviours were measured with the Opening Minds Stigma Scale for Health Care Providers in PC centres and with the Beliefs and Attitudes towards Mental Health Service users’ rights in MH centres. Reductions in both PC and MH professionals’ stigmatising beliefs and attitudes were found in the 1-month follow-up, although a ‘rebound effect’ at the 3-month follow up was detected. This emphasizes the importance of the continuity of the presence of anti-stigma activities and messages. Attrition rates were high, which can hamper the reliability of the results. Further follow-up studies should enquiry effects of long-term interventions aimed at reducing stigmatising beliefs and attitudes among primary care and mental health professionals using assessment systems that include the measurement of knowledge acquired and actual behavioural change.

## 1. Introduction

Although it may seem paradoxical, primary care and mental health professionals, those who maintain more direct and constant contact with people diagnosed with mental disorders, develop prejudices and discriminatory attitudes, e.g., stigma, towards people with mental health problems in a very similar way to the rest of the population [1,2,3] and people diagnosed with mental disorders themselves [4]. According to reviews looking at characteristics of these phenomena in primary care and mental health settings, stigmatisation towards people diagnosed with psychotic disorders is more intense than towards people diagnosed of anxiety or depression [5,6]. Evidence on the sociodemographic characteristics of professionals showing more stigmatizing behaviours points to higher rates among males or when interacting with the opposite sex [1], but is contradictory regarding age and experience, with some studies focused on alcohol abuse showing a positive correlation with age [7,8] and others, based on vignettes of people diagnosed with depression or schizophrenia [6,9], negative. These differences may be due to the combination of generational factors, together with others related to one’s own professional evolution, continued contact with diagnosed people and the development of selective tolerance according to the characteristics of each population [6,7,8,9]. Regardless of the profiles mentioned, stigma does lead to overshadowing of somatic diagnoses and pessimism about adherence to treatment [5,10,11]. This leads to disempowerment and increased internalised stigma amongst service users [12].

Most people diagnosed with mental disorders have experienced some form of discrimination in healthcare settings, with figures reaching 90% in large cohorts [13]. Despite decreases in these rates attributable to the multitude of anti-stigma campaigns implemented and, in general, the public debate about it around the world [14], levels of discrimination still prevent affected people from seeking professional help and adequate treatment when they reach health services [5,10,15,16]. However, health professionals are also key to change, since the way in which they conceptualize mental disorders ends up influencing the rest of the population [17]. Despite these facts, the systematic inclusion of anti-stigma contents in initial or continued health professional training or the implementation of specific awareness campaigns is a relatively new phenomenon [1,2]. Therefore, interventions to reduce stigma towards people diagnosed with mental disorders specifically targeted towards health professionals are being promoted [1]. Ingredients such as information about mental disorders, development of specific skills, and modifications in the culture of the health centres have been pointed out as the most effective [18].

According to recent reviews and meta-analyses [1,14,19,20,21,22,23], studies examining mental health stigma reduction interventions carried out with healthcare professionals have generally found positive results regarding interventions including social contact with people diagnosed of a mental illness and problem-based learning. However, positive results do not usually last, since the differences with control groups disappear after relatively short periods of time [21]. Furthermore, until recently, most studies have been focused on students [20,24,25] or a specific diagnostic characteristic such as people diagnosed with substance abuse [19] or borderline personality disorder [1].

However, there are three notable exceptions to the paucity of interventions applicable to different health professionals and mental health problems. The Canadian campaign *Opening Minds* applied a grounded theory model [26] to design a guide of recommendations to implement anti-stigma interventions in primary care. Following these recommendations, different organisations implemented a series of interventions throughout the geography of this country. Although there was some freedom in the format of the interventions, given the vast experience of the recovery and anti-stigma movements in Canada, all used a unified assessment methodology with a unique measurement instrument, the Opening Minds Stigma Scale for Health Care Providers [27]. A pooled analysis of their impact has shown statistically significant decreases, especially for younger professionals and those with higher initial levels of stigma [28]. Additionally, through a qualitative evaluation, the campaign has produced a synthesis of the six structural ingredients key to the success of such interventions. Programs including a recovery emphasis, personal testimony from a trained speaker who has lived experience of mental illness, employing multiple forms of social contact, teaching skills involving what to say and what to do, that employed myth-busting, and that used an enthusiastic facilitator, performed significantly better than programs that did not include all these ingredients [29].

Time to change in England carried out a large trial of a short educational intervention with medical students [30], that offered positive results. However, as the aforementioned reviews have shown [21], the results were not sustained over time. They have also released the results of a pilot project carried out with mental health professionals [31]. Multi-wave surveys carried among professionals who participated in the pilot social contact workshops carried by mental health activists (people with experience of mental suffering who collaborate with campaigns against stigma) were compared with a control group. Results showed increased understanding and confidence attitudes among professionals who participated in the workshops. 

More recently [32] the protocol of an integrative programme to reduce stigma in Chilean healthcare workers from 11 primary care centres has been published. The intervention is based on previous qualitative analyses carried out with service users and health teams [18]. The project consists of education strategies, direct (facilitators will have a diagnosis of mental disorder), and indirect contact (videos and narratives) with people diagnosed with severe mental disorders, and skill development.

As discussed above, to date there have been few national-wide campaigns, encompassing different types of healthcare professionals and possible mental health conditions, with available evaluations of interventions in healthcare settings. This article deals with the implementation and evaluation of awareness-raising interventions, aimed at reducing stigmatising beliefs and attitudes among primary care and mental health professionals. This is the first study to assess the effect of a homogeneous intervention applied to a whole territory under an autonomous health authority encompassing both primary care and mental health centres, as well as content applicable to any type of mental disorder.

## 2. Materials and Methods

### 2.1. Co-Creation Process

During 2017 and 2018, there were a series of meetings to prepare this project. Representatives of the main primary care (PC) and mental health (MH) providers in Catalonia expressed their views on the implementation of awareness-raising interventions in their centres together with Obertament technicians and activists with lived experience of mental health diagnosis and service use. There was an independent organization in charge of the moderation of this co-creation process, Spora Synergies (http://www.spora.ws/), which facilitated and offered a qualitative analysis of the meetings that allowed decision making. The contents of these analyses have not been included as it would be beyond the scope of this article. The agreements reached when reflecting on these meetings were that a local referent (a professional in charge of supervising the coordination with Obertament) was necessary for the development of awareness interventions. In the same way, it was deemed necessary to combine a theoretical part (that professionals requested to understand the need for the intervention) with first-person accounts. Additionally, it was concluded that, for the message to be truly internalised, all primary health centres must develop self-diagnosis activities that can be used to envision actions against stigma in an autonomous way, but with the accompaniment of the activists who have carried out the intervention. Two activists, Obertament collaborators with experience of extreme psychic suffering, were recruited to perform the interventions.

### 2.2. Awareness Interventions

The interventions aimed to raise awareness toward the discrimination faced by people diagnosed with a mental disorder when they use primary care and mental health services. The methodology was based on the Targeted, Local, Credible, Continuous Contact (TLC3) principles as described by Corrigan [33]. The content of the awareness-raising interventions included both theoretical and practical contents, aimed at improving professional care, promoting the participation of mental health users in decisions related to their treatment and the exercise of their rights; always combining the view of service users with the therapeutic framework in which professionals operate. The awareness-raising interventions consisted of four parts: a 4-h training workshop, a 4-h self-diagnosis workshop, a self-organised activity, and a follow-up session. Figure 1 shows a flow diagram of the intervention structure. The contents of each session are detailed below.

#### 2.2.1. Training Workshop

In this part, pedagogical contents were combined with different Obertament videos (indirect contact) that had been used during the social-marketing campaigns (https://obertament.org/ca/sanitat/per-que-sanitat) as well as first-person accounts (direct contact) of how activists have coped with mental ill-health and how discrimination affects recovery outside and inside the healthcare system. Participants were invited to ask questions and were provided with information on peer- and relatives support organizations. The contact component of the intervention emphasised evidential aspects about different recovery journeys and tried to refute myths about mental illness. The pedagogical contents of the session follow.

##### First Block: Mental Health and Mental Illness

After a presentation, professionals were invited to participate in a dynamic group conversation to estimate and discuss the prevalence of mental health problems. The opinions of the participants were contrasted with the fact that, according to the World Health Organization, one in four people will have a mental health disorder during their lives. Therefore, it became patent that mental disorders are not exceptions but commonplaces. Next, activists gave a lecture combined with personal accounts focused on the following components: mental health as more than mental illness, mental health in the community, social determinants of mental health, myths about mental health problems, mental crises as an exception and not a rule in the lives of diagnosed persons. After a break, the concepts of recovery and discrimination were discussed also using personal accounts. The key messages were: (1) recovering a purpose in life is different from being ‘cured’ and (2) a good part of distress experienced by persons diagnosed with mental health disorders is not due to the symptoms inherent to their disorder but to the discrimination they suffer because of being identified as having a diagnosis of a mental disorder. This happens by making public that a diagnosis has been received or simply by involuntarily showing the side effects of medications.

##### Second Block: Stigma and Discrimination

The block began with a lecture on the concept of stigma that had the following components: Definition of stigma and categorization, stigma and prejudices, prejudices in mental health. The objective was for participants to be introduced to the concept of stigma and to become aware of how human beings need to label behaviours and social groups to feel safe. It was explained that, although this has an evident adaptive role, when we label disadvantaged groups, we tend to justify the discrimination to which they are subjected. This is a process with varying levels of awareness. Therefore, it is important to be able to make our feelings conscious analysing the prejudices we have towards other people and groups. This allows health professionals to make decisions adapted to each situation and therefore less biased.

Once the concept and scope of stigma had already been clarified, the lecture followed by explaining and exemplifying how stigma causes discrimination and in which domains, including health contexts. For example, it was explained that many people think unjustifiably that people with a mental disorder cannot study or work. This makes it less likely that a person with this condition can complete a course or get a job, reinforcing the prejudices. In this way, both diagnosed people (by not trying), employers (by negatively evaluating this characteristic in a selection process) and society as a whole (by taking for granted the fact that people diagnosed cannot achieve certain goals) contribute to fulfilling this unfounded prophecy.

##### Third Block: Healthcare and Primary Care/Mental Health Care

This part had some different components depending on whether it was implemented in PC or MH health centres. The block commenced with a discussion on the differences between ‘personal’ and ‘professional’ treatment. This discussion was contrasted with examples and personal experiences of the activists. The message was that although there are available professional-quality treatments, sometimes service users perceive that the personal treatment they receive does not help them to recover. Next, data about treatments and care load adapted to PC and SM contexts was offered. This was contrasted with the data available on discrimination suffered by service users at both PC and SM facilities.

##### Fourth Block: Fighting Stigma

In this final part, activists briefly described the Obertament project and its objectives as well as similar initiatives around the world. To reinforce the legitimacy of the activity, special attention was paid to other projects that have developed similar interventions targeted to health professionals in other countries such as Canada and the United Kingdom (see Section 1). Before finishing, the implementation referent (a professional from each PC or MH centre), explained how the selected self-organised activity would be carried out.

#### 2.2.2. Self-Diagnosis Session

The second session was focused on a self-diagnosis carried out by the PC teams in relation to stigma and discrimination. That is, it aimed to give tools to professionals so themselves can define jointly what stigmatizing practices they detect in their own work environment and how they could reverse them. A group dynamic was used with three main objectives: (1) Identifying the stigmatizing and discriminatory practices and behaviours that take place in their own working centre and (2) defining proposals agreed between the team to deal with the practices and behaviours identified as stigmatizing and (3) prioritizing the proposals and plan their implementation. This session was not implemented with the MH teams for reasons of training time availability.

#### 2.2.3. Follow-Up Session

This shorter session, carried out approximately a month after the baseline assessment, was used to reinforce the concepts learned in addition to sharing the progress of the self-organised activities. When these activities included a public presentation, this was done during this session.

### 2.3. Procedure

The anti-stigma awareness-raising interventions were evaluated through a prospective single-blind cluster-randomised-controlled trial. Each cluster was a health centre. The design was single blinded since the evaluators did not know which centres have been evaluated as case or control. Only the organisers of the intervention workshops had this information. The twelve centres (6 PC + 6 MH: 2 adult MH, 2 children and adolescent MH and 2 substance abuse) taking part in the study were designated by the Department of Health of the Catalan Government blindly to the evaluators before randomisation.

Upon enrolment, each centre was included in a randomisation table with a fixed number of control (3 PC + 3 MH) and intervention (3 PC + 3 MH) sites. In this way, each centre had the same possibilities of belonging to the experimental or to the control conditions. Once the centre was randomised, professionals received a registration questionnaire, which included the baseline assessment. It was composed by sociodemographic (including gender, age, and personal contact with mental disorders) and professional (profession category) information + a tailored beliefs and attitudes sale (see below Section 2.5). The centres included in the experimental group accessed the course immediately, while the centres included in the control group waited, giving time to carry out the follow-up assessments within the intervention group, before beginning the course. Both groups completed a follow-up at one month and another at three months after the baseline assessment. Figure 2 shows a flow diagram of the design.

### 2.4. Participants 

Considering as possible universes all primary care and mental health centres (children and adolescents, adults and substance abuse) in Catalonia and the effect sizes of an intervention carried out with social work students [34], using a similar methodology, the number of participants was estimated according to the following calculation of statistical power. Accepting an alpha risk of 0.05 and a beta risk of 0.2 in a two-sided test, assuming a correlation between the first and second measure of *r* = 0.6, 99 subjects were considered necessary in each group to recognize as statistically significant difference greater than or equal to 0.4 standard deviations.

Following these sample size calculations, the intervention workshops were implemented in randomly assigned Catalan PC and MH centres comprising a total of 371 professionals (185 PC + 186 MH). The recipients were professionals working in such settings: administrative officers, general practitioners, odontologists, nurses, psychiatrists, psychologists, and social workers. Figure 3 and Figure 4 show flow diagrams of the recruitment and follow-up process.

### 2.5. Instruments

The Opening Minds Stigma Scale for Health Care Providers (OMS-HC; [27,35]), is a 15-item scale that measures stigma levels and has been specially designed for use in primary care. The overall internal consistency for the whole scale (α = 0.79) and three subscales, namely Attitudes (α = 0.68), Disclosure (α = 0.67) and Social distance (α = 0.68) has been found to be acceptable. Internal consistency is also satisfactory across different health professional groups (physicians, nurses, etc.). The scale has been successful in detecting positive changes (SRM ≤ 0.50 to ≤ 0.91) in various anti-stigma interventions [35]. The Attitudes subscale includes different attitudes of health care providers towards people with mental illness such as being comfortable helping a person who has a mental illness, having negative reactions, feeling that there is little to do, thinking that people with mental illness do not try hard enough to get better or feeling compassion. The Disclosure subscale includes items on their willingness to seek help and disclose if they were in treatment for a mental illness to colleagues or friends. Finally, the Social Distance subscale includes items on willingness to work with colleagues who had a managed mental illness, going to a physician if they knew that the physician had been treated for a mental illness or wanting a person with a mental illness, even if it were appropriately managed, to work with children or living next to them.

The professionals’ Beliefs and Attitudes towards Mental Health Service users’ rights scale (BAMHS; [36]) has been developed by our group for this project. In preliminary analyses carried out within cross-sectional studies, we have found four subscales: justification beliefs (α = 0.70), coercion (α = 0.65), paternalism (α = 0.71) and discrimination (α = 0.65) with good global reliability (α = 0.87). The justification beliefs subscale materialises the professional beliefs that health-related professionals have which justify the status quo including items claiming that mental disorders are diseases like any other, that aggressiveness is due to mental disorders, that it is not possible to recover without the intervention of a professional, and that some patients will never recover. The coercion dimension addresses recurrent topics with mental health professionals when discussing service users’ rights including questions on involuntary hospitalization, mechanical restraints, and, inversely, respect for service users’ autonomy. The paternalism subscale represents a series of beliefs related to the supposed inability of people diagnosed with mental disorders to take charge of their lives including having children, making decisions regarding their treatment, or prioritizing treatment over dignity. Finally, the discrimination subscale materialises widespread prejudices towards mental health service users such as voting rights, friendship, overuse of emergency settings or feeling comfortable if a person with a mental disorder were a teacher in a school.

### 2.6. Statistical Analyses

Instrument reliability was measured using Cronbach’s alpha. Baseline comparability between groups (including sociodemographic and professional data and scale scores) was assessed using Odds ratios and χ^2^ tests for categorical data and Student’s *t*-test for continuous data. Attrition was analysed by means of Mann–Whitney’s U or *t*-tests comparing baseline characteristics and scores as well as differential scores between baseline and the first follow up between participants who completed the second follow-up and those not completing it. All the participants were included in the analyses in the group to which they were randomised irrespective of whether they have missing data. To evaluate the differences between groups at the follow-up session (1 month after baseline) and the 3-months follow-up assessment, depending on normality of distributions and available data, we used Mann–Whitney’s U or independent measures *t*-tests and Wilcoxon signed-rank or repeated measures *t*-tests (the latter in each group separately) and general linear models considering the evolution of intervention and control groups. Due to the high attrition rate, we used multiple imputations with chained equations to account for missing information. We carried sub-analyses for the adult, children and adolescent, and substance abuse mental health centres. Additionally, we used multilevel mixed-effects linear models. 

## 3. Results

### 3.1. Primary Care

The reliability of the stigma measure (OMS-HC) was acceptable staying above *α* = 0.7 for the instrument as a whole and above *α* = 0.5 for all subscales (attitudes, disclosure, and social distance), except for the disclosure subscale in the second follow-up (*α* = 0.4).

Regarding baseline characteristics (Table 1), there was a statistically significant difference between the intervention and control groups in the total score of the OMS-HC scale (*t* = 2.138, *p* < 0.05), although not in any of the subscales. There were also differences between the control and intervention group in terms of gender (67% vs. 82%; *χ^2^* = 5.223, *p* ≤ 0.05). No gender differences were found for any of the OMS-HC subscales. A mild statistically significant correlation was found between the Social distance subscale and age at baseline (*r* = 0.172, *p* < 0.05). Attrition analyses did not show any baseline differences between completers and participants lost to follow up, but it detected that participants who responded to the second follow-up had had greater reductions in the total stigma score (*z* = −2.459, *p* < 0.05).

No statistically significant differences were observed between intervention and control group scores at any of the follow-up points (Table 2). Repeated measures *t*-tests showed no statistically significant change in the control group while in the intervention group statistically significant decreases were seen between baseline and first follow-up for the OMS-HC total score (*t* = 2.813, *p* < 0.01) and the disclosure subscale (*t* = 2.534, *p* < 0.05). There was not enough sample to make comparisons between the first and second follow-up without multiple imputations. When performing multiple imputations to recover lost data, similar results were observed between baseline and first follow-up. This allowed us to estimate that differences in evolution between first and second follow-up were statistically significant (downward in the intervention group) for the total score, and the disclosure, and social distance subscales.

The general linear models showed a statistically significant drop between the first observation and the second for the OMS-HC disclosure scores with statistically significant effects (*F* = 26.881, *p* < 0.001) on the quadratic evolution, which implies a ‘rebound effect’ (there was a very marked drop in the intervention group which then returned to higher values). These results for the disclosure subscale were generally confirmed and expanded to the OMS-HC total score using imputed data. Finally, multilevel mixed-effects linear models showed no effects for any of the scores.

### 3.2. Mental Health

The reliability of the beliefs and attitudes measure (BAMHS) was acceptable, remaining above *α* = 0.7 for the instrument as a whole and above *α* = 0.4 for all subscales (beliefs, coercion, paternalism, and discrimination), except for the disclosure subscale in the second follow-up (*α* = 0.16). 

Regarding baseline characteristics, there were no statistically significant differences in any sociodemographic characteristics or in the scores of the questionnaire (Table 3). No gender differences were found for any of the BAMHS subscales or total score. A mild statistically significant correlation was found between the Paternalism subscale and age at baseline (*r* = 0.209, *p* < 0.005). Attrition analyses did not show any baseline or differential scores differences between completers and participants lost to follow up.

Statistically significant differences were observed between intervention and control group for beliefs (*z* = −2.419, *p* < 0.05) and the total BAMHS score (*z* = −2.392, *p* < 0.05) at the first follow-up point (Table 4) but no differences were observed in the second follow-up. The results of the repeated measures *t*-tests showed a statistically significant decrease in the control group of the discrimination subscale between the first and second follow-up. In the intervention group there was a trend in the decrease of the total score (*t* = 1.708, *p* = 0.091) and a statistically significant decrease of the coercion subscale (*t* = 3.056, *p* < 0.005) between the baseline and the first follow-up. When we carried out calculations by type of centre, we saw that these results corresponded mainly to adult mental health centres (where both total and coercion had statistically significant decreases between baseline and first follow-up). In the case of children and adolescent centres and addiction centres, the sample size only allowed comparisons between baseline and first follow-up. In the case of children and adolescent centres, the intervention group had a statistically significant decrease between baseline and first follow-up in the total score and in beliefs about the system (which reflects an increase in self-criticism). In the case of substance abuse, no statistically significant changes were found in any of the samples or differences in evolution. General linear models showed different evolutions just in the case of the discrimination subscale (*F* = 5.450, *p* < 0.01).

Calculations carried out with imputed data clearly confirmed the results on coercion although, as was the case in the primary care, there was a ‘rebound effect’. In addition, general linear models showed differences in overall evolution between the control and intervention groups in the discrimination scores, with decreases in the intervention group in the first follow-up, which are then softened in the second in most of the imputed scenarios. Finally, multilevel mixed-effects linear models showed statistically different evolutions in the total (*F* = 3.922, *p* < 0.05) and beliefs (*F* = 5.277, *p* < 0.05) scores.

## 4. Discussion

This study aimed to evaluate two awareness-raising interventions respectively tailored to reduce stigma beliefs and attitudes towards persons with a mental health diagnosis among primary care and mental health professionals. The results show intervention size effects close to those registered in other territories where similar activities have taken place [1,14,19,20,21,22,23]. Despite these generally favourable results, as seen in the previous literature [21] a rebound effect is seen in further follow-up assessments. We hypothesise that this may be due to the short duration of the interventions and the fact that the first follow-up assessment of the intervention groups was done after a session of self-organised activities with a high degree of motivation on the part of participants. This emphasizes the importance of the continuity of the presence of the anti-stigma movement through reinforcement activities. Therefore, to implement and consolidate a cultural change like the one proposed with these interventions, it is necessary to continue working in this line. Such changes require time and a strong alliance between the organisations involved and the public administrations. In this sense, the project intends to be an active agent of change, promoting awareness and changing attitudes to advocate for the rights of people with mental health problems in healthcare contexts. 

The results of this study and, in general, the efforts to implement this type of awareness-raising projects against the stigma on people diagnosed with mental disorders, should be seen in the context of the struggle for a more inclusive society in the new era opened by the Convention on the Rights of Persons with Disabilities [37] in which health it is considered a right for all. As commented in the introduction, stigma in the field of mental health care causes delays and resistance to the use of services that contribute to the aggravation of mental health conditions [2]. At the primary care level, many people do not get the treatment they need since many health professionals attribute somatic symptoms to mental health problems. Cases have been reported in which this ‘diagnostic overshadowing’ has caused that heart attacks could not be detected correctly, or transplantations have been delayed or even denied [28]. In the case of mental health, many delay help seeking due to fear of discrimination [13]. The results of this project will serve to justify anti-stigma awareness-raising projects carried out with healthcare trainees and professionals at the local and international levels that can prevent these situations from occurring, and therefore improve the physical and mental health status of people diagnosed with mental disorders. This is a very important aspect in a context of scarce resources.

Among the limitations of this study, the attrition rate is particularly remarkable. Although it is somewhat common in this type of interventions [38], in which prejudices about service users are disclosed, discussion on this issue is needed. Some professionals failed to fill the baseline questionnaires, especially in the case of the control groups. This caused the number of participants in both groups to be unbalanced. Additionally, problems carrying out follow-ups, included the high healthcare pressure of professionals and several strikes and civil servant evaluations that coincided with the implementation of this study. This situation may create doubts about the reliability of this study results, especially among primary care professionals in whom we see a higher response rate among participants with higher initial effects, that must be resolved with further follow-up studies. Additionally, formative evaluation activities were not carried out due to lack of time and resources. Future studies should include evaluation measures to quantify the degree to which participants grasped the contents of training actions. Finally, only beliefs and attitudes were measured, but not actual behaviours. Future evaluations of these types of interventions carried out with more resources should assess whether they have a direct impact on the actual health professionals practice.

## 5. Conclusions

This project aimed at raising awareness on the stigma experienced by people with mental health problems in the healthcare field by trying to promote reflection on clinical practice among healthcare professionals. We intended to foster a change of attitudes and practices in relation to the need to guarantee the rights of service users. Although the results of the evaluation of the first implementation at the Catalan level point to favourable but mild and ephemeral effects, interventions with long-term follow-up activities must be implemented and evaluated using assessment systems that include the measurement of knowledge acquired and actual behavioural change to ensure the impact of the anti-stigma movement in healthcare practise.

## Figures and Tables

**Figure 1 ijerph-18-01214-f001:**
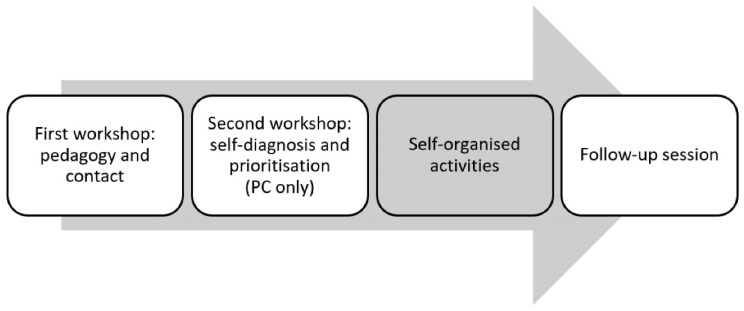
Flow diagram of the intervention structure. The sessions supported by the activists are coloured in white, the activity self-organised by the professionals is coloured in grey. PC: Primary Care.

**Figure 2 ijerph-18-01214-f002:**
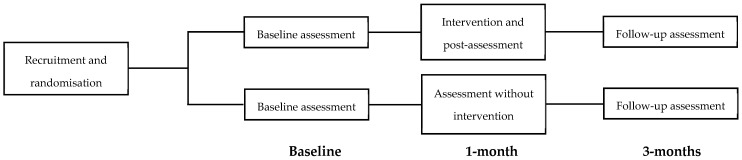
Flow diagram of both evaluation designs.

**Figure 3 ijerph-18-01214-f003:**
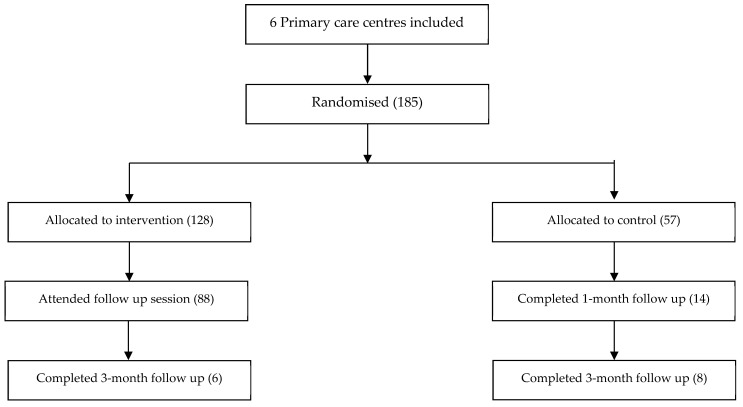
Flow diagram of the primary care centres recruitment and follow-up process.

**Figure 4 ijerph-18-01214-f004:**
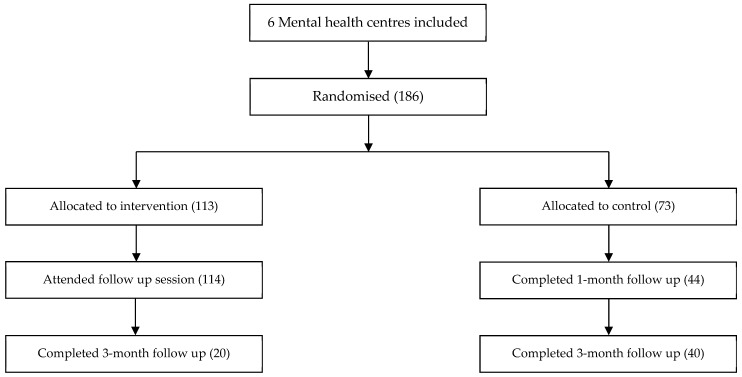
Flow diagram of the mental health centres recruitment and follow-up process.

**Table 1 ijerph-18-01214-t001:** Sociodemographic characteristics and baseline scores of primary care participants.

	All (*n* = 185)		Intervention (*n* = 128)		Control (*n* = 57)		Statistical Significance		
	N	%	N	%	N	%	OR, 95% CI		*p*
Gender (% females)	137	70.6	112	81.8	38	66.7	2.24. 1.11–4.52		0.022
Experience with mental health issues (%)	80	41.7	52	38.5	28	49.1	0.649. 0.35–1.21		0.173
Relative with mental health issues (%)	135	69.2	96	69.6	39	68.4	1.055. 0.54–2.05		0.875
	M	SD	M	SD	M	SD	t	*p*	d
Age (M ± SD)	44.34	10.82	44.02	11.41	45.05	9.41	0.630.	0.530	0.095
Attitudes S	2.20	0.57	2.25	0.58	2.11	0.52	1.570	0.118	0.0897
Disclosure	2.50	0.68	2.55	0.67	2.39	0.69	1.504	0.134	0.1073
Social distance	2.12	0.59	2.16	0.62	2.05	0.51	1.171	0.243	0.0938
OMS-HC total	2.36	0.40	2.40	0.41	2.26	0.37	2.138	0.034	0.0626

**Table 2 ijerph-18-01214-t002:** Follow-up scores among primary care participants (raw data *).

	First Follow-Up				Second Follow-Up				Statistical Significance								
	Intervention		Control		Intervention		Control		First Follow-Up			Second Follow-Up			RM-ANOVA **		
	M	SD	M	SD	M	SD	M	SD	z	*p*	d	z	*p*	d	F	*p*	ηp^2^
Attitudes S	2.15	0.60	2.21	0.50	2.22	0.61	2.25	0.53	−0.566	0.571	−0.115	−0.521	0.602	−0.057	1.708	0.259	0.363
Disclosure	2.43	0.74	2.43	0.40	2.17	0.38	2.66	0.60	−0.015	0.988	−0.003	−1.511	0.131	−0.948	11.878	0.008	0.798
Social distance	2.04	0.63	2.24	0.45	1.93	0.37	2.13	0.56	−1.197	0.231	−0.332	−0.393	0.694	−0.393	0.631	0.564	0.174
OMS-HC total	2.26	0.48	2.38	0.30	2.18	0.31	2.41	0.43	−0.725	0.468	−0.249	−1.035	0.301	−0.587	1.911	0.228	0.389

* Data before applying multiple imputation; ** Group × time interaction within 3 time points.

**Table 3 ijerph-18-01214-t003:** Sociodemographic characteristics and baseline scores of mental health participants.

	All (*n* = 186)		Intervention (*n* = 113)		Control (*n* = 73)		Statistical Significance		
	N	%	N	%	N	%	OR, 95% CI		*p*
Gender (% females)	177	78.3	105	76.6	72	80.9	1.291. 0.667–4.498		0.448
Experience with mental health issues (%)	87	38.5	49	35.5	38	43.2	1.380. 0.799–2.386		0.248
Relative with mental health issues (%)	174	78.7	106	78.5	68	79.1	1.034. 0.533–2.004		0.922
	M	SD	M	SD	M	SD	t	*p*	d
Age (M ± SD)	41.56	12.35	41.72	12.70	41.30	11.81	−0.237	0.813	0.033
Beliefs	2.46	0.39	2.44	0.39	2.49	0.39	0.884	0.378	−0.133
Coercion	2.48	0.46	2.49	0.44	2.46	0.49	0.881	0.38	0.070
Paternalism	2.30	0.40	2.27	0.39	2.34	0.43	−0.467	0.641	−0.179
Discrimination	1.91	0.42	1.93	0.45	1.88	0.38	−0.456	0.649	0.120
BAMHS total	2.36	0.27	2.35	0.27	2.39	0.28	0.910	0.364	−0.137

**Table 4 ijerph-18-01214-t004:** Follow-up scores among mental health participants (raw data *).

	First Follow-Up				Second Follow-Up				Statistical Significance								
	Intervention		Control		Intervention		Control		First Follow-Up			Second Follow-Up			RM-ANOVA **		
	M	SD	M	SD	M	SD	M	SD	z	p	d	z	p	d	F	p	ηp^2^
Beliefs	2.41	0.40	2.55	0.37	2.52	0.35	2.56	0.37	−2.419	0.016	−0.349	−0.166	0.868	−0.349	0.388	0.681	0.019
Coercion	2.38	0.42	2.47	0.49	2.42	0.39	2.43	0.48	−1.449	0.147	−0.195	−0.360	0.719	−0.195	0.478	0.624	0.023
Paternalism	2.28	0.43	2.32	0.40	2.23	0.39	2.32	0.40	−1.006	0.314	−0.093	−0.695	0.487	−0.093	0.065	0.937	0.003
Discrimination	1.94	0.45	1.88	0.46	1.88	0.38	2.06	0.39	−0.594	0.552	0.130	−1.549	0.121	0.130	5.450	0.008	0.214
BAMHS total	2.33	0.29	2.39	0.29	2.33	0.28	2.41	0.27	−2.392	0.017	−0.235	−0.700	0.484	−0.235	0.781	0.465	0.038

* Data before applying multiple imputation ** Group × time interaction within three time points.

## Data Availability

The databases used for this study are available online as Appendix A of this article.

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
