# Peer review of "Efficacy of an Intervention to Reduce Stigma Beliefs and Attitudes among Primary Care and Mental Health Professionals: Two Cluster Randomised-Controlled Trials"

_ijerph, 2021, doi:10.3390/ijerph18031214_

Round 1

Reviewer 1 Report

Dear Authors
The paper presents a good theme, current in these times. Some observations are attached in the attachment:
Regarding the Introduction:
1. There is more information about the subject that can be oriented to the need to carry out this study. For example, DOI: Vaccari et al., 2020). 10.1186 / s13033-020-0340-5. eCollection 2020; Sapag et al., 2018). DOI: 10.1080 / 17441692.2017.1356347; Daumerie et al., 2012. DOI: 10.1016 / j.encep.2011.06.007, as well as other recent publications on WoS.
2. This writing requires some citations to support the statement, relevant to this study, especially regarding the results of similar experiences, recommendations, of which there are publications in WoS, for example, Ori et al., 2020. DOI: 10.1186 / s12888-020-02902-8.
3. Are the ingredients that accompany anti-stigma actions similar in all the incursions that have been carried out? Which is the most important or do they all converge equally to the referred antistigma actions? The role of awareness-raising has been addressed in otherspapers, why is that point that is essential in your proposal not discussed?

4. Regarding the Methodology:

The initial explanation of the methodology is confusing. It can be summed up in the fundamentals. Perhaps first point out that it is a mixed methodology study. There was an initial qualitative procedure and then a verification of quantitative data of some variables observed in the qualitative procedure. Next, who are the participants and what role does each of them play. Agents who must be defined (health personnel, users. It is not explained that he is an activist, then a group of participants, professionals, administrators, social workers, among others) attended. Here the statistical processing is pertinent to designate the population sample evaluated. A table would help this, in addition to establishing the difference between the intervened group versus the control group.

5. Some questions: Did the intervention with workshops collect data or was it educational only? If so, how do you ensure that there was learning, or what you call awareness in the intervention? A baseline is mentioned, how was it measured? where is the data to be compared? Also, how they evaluated post session?

6. Then, before describing the procedure, clarify the characteristics of the instrument (s) used, both qualitative and quantitative. Likewise, the cut-off points, for later interpretation.

7. Once the above has been detailed, the methodology in general, participants, instruments, the description of the procedure proceeds, to complete with the type of statistical analysis. The mixture of this makes understanding difficult.

8. In the instruments as in the procedure, it is convenient to refer to more previous studies carried out to ensure that they are instruments that are justified in this study. They are apparently well-chosen, but they should include further WoS references.

9. Regarding the Results
The results show the statistical data of OMS-HC and BAMHS. Report the reliability statistics well, although the results are discrete in general, even more so in the subscales. It is desirable to have the data for the calculation of T, displayed in a comparative table.

10. In addition, the graphs in Figures 5 and 6 should contain the error bars, so that there is greater clarity of the findings.

11. The same situation for the ANOVA results of OMS-HC and BAMHS, to indicate the statistics that explain F and p shown, to interpret the statistical significance. In this case, a post hoc analysis is missing.

12. The discussion describes the main results and, in some statements, supports them with some references. However, it is not so obvious that the awareness program was the cause, as reported above. It is also shown with incomplete data for statistical demonstration and interpretation, as to ensure a "positive result". The explanation is not based on the research findings and raises a procedural problem. On the one hand, it is necessary to be careful in the findings. For example, "the evidence suggests that ...". The scant evidence reported by the program may even invalidate the data obtained. On the other hand, it is attributed to a lack of reinforcement, when the activities seemed to be under another psychoeducational context, and at least none supported by behavioral theory. There is no report on the people who dropped out of the program. Although it is explained with external factors, the importance of a strategy of this nature should overcome any obstacle. At least, if it is pointed out as a weakness.

13. The conclusion mentions that the project or study has generated a reflection. If this is framed in the approach of future problems and challenges around the subject, it could be treated as a good discussion. The work leaves questions yes. It raises topics already covered in other latitudes and that are currently in the chosen Catalan context. Which marks a line of investigation that is worthy of note. The care that needs to be attended is the same as Henderson et al. (2014), which refers to the research regarding the sensitivity of the instruments, social desirability, the language used, among others (also pointed out by Eiroa-Orosa et al., 2019). Therefore, it is not pertinent to be so categorical with the results. He also points out the relativity of the results over long periods in Gronholm et al., (2017). Finally, point out 3 references that are reviewed to state that similar results are obtained. The care in this, to explain the validity of the result, is the data that is used in the statistics to justify significance, not necessarily the report of results without said empirical precision.
Finally, the bibliographic references show little research with close empirical results that can be compared with those obtained in this study.

14. Spelling checks regarding language are indicated in the text

Author Response

Please see the attachment. Thank you in advance.

Reviewer 2 Report

This paper evaluates the effectiveness of an intervention to improve the attitudes and beliefs of health personnel towards mental health problems. The field work seems rigorous and is a topic of interest to readers.

I have some concerns, mainly about methodology.

  1. Introduction.

The introduction is correct, but brief. There is scope to develop some ideas that serve as a starting point for the reader and help to contextualize the research well.

For example:

1.1. There is a remarkable amount of literature studying attitudes of physicians towards people with mental illness (see for example the review by Vistorte et al., 2018). It would be interesting to mention some of these studies to contextualize the scope of the problem.

Vistorte, A. O. R., Ribeiro, W. S., Jaen, D., Jorge, M. R., Evans-Lacko, S., & Mari, J. D. J. (2018). Stigmatizing attitudes of primary care professionals towards people with mental disorders: a systematic review. The International Journal of Psychiatry in Medicine, 53, 317-338. https://doi.org/10.1177/0091217418778620

1.2. Page 1. “Older physicians tend to be more attuned to these attitudes and when these occur, there is overshadowing of the diagnosis and pessimism about adherence to treatment [3]”.

It would be interesting to explain if this is caused by initial training of older physicians; or by years of experience,…

1.3. There are also some other interventions (although on a smaller scale than those mentioned in the introduction) that could be commented in the introduction. For example:

Shen, Y., Dong, H., Fan, X., Zhang, Z., Li, L., Lv, H., ... & Guo, X. (2014). What can the medical education do for eliminating stigma and discrimination associated with mental illness among future doctors? Effect of clerkship training on Chinese students' attitudes. The International Journal of Psychiatry in Medicine, 47(3), 241-254. https://doi.org/10.2190/PM.47.3.e

  1. Participants.

The number of participants in PC was not balanced between the experimental and control groups. The experimental group had more than twice as many participants at the start of the study (maybe it also happened in MH centres).

This should be indicated in the limitations of the study.

It should be interesting to add a table indicating the sex of participants, their professional role and, if possible, some demographic information that could be interesting for the study, such as age and years of experience.

Also, it should be indicated if there were statistically significant differences in these variables between experimental and control groups (at least at the start of the study).

In figure 3, the use of the letter n before the number of participants should be unified. It is recommended to use it in all cases, not just one of them.

In figure 4 the n is missing in most cases.

The sentence "There were also differences between the control and intervention group in terms of gender (67% vs. 82%; χ2 = 5.223; p = 0.022)" (page 8), should be relocated to "participants" section.

  1. Instruments.

The content of each subscale should be briefly specified, in addition to providing its name. It would be interesting to provide an example of an item in each subscale, or, optionally, include the instruments in annexes.

  1. Results.

In addition to the p-values, the effect size should be indicated, that is, Cohen's d statistic (for the t test) and eta squared partial (for ANOVAs).

Figures 5 and 6 are difficult to analyze: both because of their size, because of the amount of information they contain; and because it is difficult to appreciate where the statistically significant differences are (this information is provided in the text, but it is difficult to appreciate it in the figures).

It is recommended to transform this figure into a table in which the means and standard deviations of the subscales for each group, the F, p and partial eta squared values ​​are shown.

On page 7 it states: "We carried sub-analyzes for the adult, chil-drain and adolescent, and substance abuse mental health centers." However, no information is provided from these sub-analyzes. It would be very interesting information.

Additionally, it would also be interesting to compare the scores on the scales of the different professional figures who participated in the study. That is, are there differences in attitudes and beliefs among administrative officers, general practitioners, odontologists, etc?

As those professionals have very different initial training and their work and relationship with the users of the centers is of a different nature, it’s possible that there are differences in attitudes and beliefs between groups of professionals.

  1. Discussion and conclusion.

These final sections are too generic. It would be interesting to briefly highlight the importance of the results, the positive effects that these interventions can have in terms of public health and improvements in the health of users. Also, it would be interesting to highlight the implications of the results in the initial training of health personnel.

The authors can even consider the social implications of the results, adding some information about the social context in which the work has been carried out and highlighting the implications of their intervention to achieve a more inclusive society.

Author Response

(The authors gave the same response as above.)

Round 2

Reviewer 1 Report

Dear Authors.

I reiterate that the study is interesting and contributes to a theme that must be addressed, especially in these times. However, take into account the observations made to clarify some key aspects:

i) The rationale for and need to address this study, based on what has been published so far, clarifying the study variables.

ii) The methodology must be clear in order that what has been done will be presented and discussed in the results. In this case, it is necessary to consider that the intervention will influence the result. For which it must be carefully exposed.

iii) The results will be better evaluated to the extent that they consider all the processed data and are also presented appropriately. Use graphs provided by statistical programs.

iv) The discussion as the conclusions do not refer to the claims, if not to the contributions or what they contribute to clarify the issue, in this case, is there a reduction of stigmatization with this procedure? Please attend to each of the suggestions.

a greeting

 Observation:

In abstract:

A disadvantage with keywords: Line 19, 33: Obertament, is an Institution (Obertament Catalan Alliance), therefore it does not correspond as a keyword, because it is not an intervening variable in the investigation. Activism is not a study variable, as is participation.

English expression, change according to the specified line:

23, 26, 27: "center" for "Center"

25: "behaviors" by "behaviors".

27: "stigmatising" by "stigmatizing"

30: "where" for "were"

31 inquiry by inquiry and stigmatizing by stigmatizing.

In Introduction:

Some suggestions that have not been clarified: The examined variable is key. Checking your writing is important. For example, I suggest: The discriminatory attitude or stigma (more current denomination) towards people diagnosed with psychotic disorders is more intense than in other diagnoses such as anxiety and depression, in mental health and primary care contexts (ref 5.6). Also ... (point out other contexts referred to in the bibliography that highlight more clearly the evidence regarding said attitude). The literature does not clearly differentiate between the terms stigmatization, prejudice, or discriminatory attitude in this context. It should be mentioned in a paragraph, to decide the term to be used in the variable examined in the study, expressing that the others will be considered as synonyms. Line 47 - 50: It obeys an opinion of the author that can be incorporated at the end of the introduction as a basis for the study carried out. Line 51: Add: Regardless of the profiles mentioned. In paragraph 54-67, it requires clarification: what is remarkable on the one hand is that anti-stigma actions have been carried out; point out. On the other hand, it has been insufficient due to the evidence that people affected with this type of diagnosis resort less to health centers. This point should be complemented with the following paragraph (68-75), in pointing out the discrete results in different initiatives, to expose the problem based on the need to develop a more effective procedure (as the title indicates) to address stigmatization, despite what was implemented based on the evidence shown in paragraphs 76-105. In this, it is advisable to clarify the weaknesses of the other initiatives that motivated this study. What is expressed in lines 76-78 should be at the end, within the foundation of the problem. Line 78-93: As it refers to a single study, it should be summarized in the main. Line 82: avoid the adjective "large". Line 94: avoid adjective "large" Line 95: If the fact that it was "without endurance" is relevant, explain it. If not, delete it. Line 102-107: Come refer to a single study, summarize it fundamentally in a few lines. Line 108-111. This paragraph should be complemented with the evidenced justification of the weaknesses of what has been done in other contexts, which justifies the need for its realization.

Line 111. As indicated in the methodology, the participants are primary health professionals and Health Centers. It is not relevant who carried it out, but rather the evidence of the findings, regardless of the participation of the protagonists of the research. It may even imply a bias, due to the fact that two of the authors belong to the aforementioned Organization.

In Materials and Methods:

In the co-creation process, without technically specifying the methodology used (qualitative?), Conclusions mentioned here, are obtained that are used in the research. If this is relevant, why is it not included? They are also directed by 2 Obertament activists: Again the meaning of this expression "activist" is not specified, referring to people with lived experience in mental health services. Regarding awareness, if you follow what Corrigan et al., 2014, it would be enough to mention its principles, which are the 3 indicated in the conclusions of said paper. The steps outlined should be consistent with it. Therefore, it should refer that the indicated workshops correspond to the indicated appointment or another or with modifications unless it is of own elaboration. This aspect should be specified briefly.

The universe and the selection of the participating sample are not clearly specified (line 247).

Line 143 diagram: change self-organised for self-organized.

Line 154: change "emphasis and tried to refute myths" to "emphasis evidential aspects about ...". From the first workshop, blocks 1 and 2 should focus on methodological aspects rather than on emphasizing results. For example 158-161; 166-171 (It is not clear if it is part of the contents of the methodology used). Line 172: In the second block, the subtitle stigma and discrimination is used. They are different? Defining the precise term in this variable has already been mentioned. Line 177-189: They should refer to some sentences that reflect an opinion that tries to justify the procedure. On the other hand, it refers that "it was clear" (line 182), which indicates that it is a result. It is not convenient to indicate it in the methodology. This is pertinently indicated in the third and fourth blocks, for which there are no objections to them.

Line 173: Change "on stigma" to "the stigma"; 176: "level behaviour" to "level behaviors"; 179: "conscious analysing" for "conscious by analyzing"; 182: "was clear" for "were clear". 188: " and the society as a whole" for "and society as a whole"; 192: "centres" for "centers"; 192: "of" for "between"; 195: "professional quality" for "professional-quality".

See the words in the manuscript and correct them.

Line 215: There are not specified the step.

line 217: change "approximately month after" for "approximately a month after"

In Procedure: See comas and words mentioned in the text.

Line 227: When were criteria designed for the Centers of health?

line 229: change randomisation for randomization.

Line 240-242: This comment no applies, Correspond to the result. The ethics aspect have to be explicated in other apart.

In figure 2, the image is blurry. Line 244, Remove the 3-month indication from inside the rectangle.

In Participants:

Change in line 248: Considering effect for Considering the effect.

In the Instruments and Statistical Analyses:

line 266: "change "is an 15-item scale" for "is a 15-item scale"; 297: "subscale materialises widespread" for "subscale materializes widespread"; "t test" for "t-test".

In result.

Review commas and previously mentioned words.

The results are shown in tables. It was recommended to show comparative graphs that clarify the presentation.
There is an incompatibility with what is written and presented in the tables. In tables 2 and 4 it is mentioned in the title that they correspond to raw data (lines 385 and 391). In the text, it is also mentioned that they correspond to total data (lines 322 and 352). However the tables present M and SD data (supposedly mean and standard deviation) (Lines 383, 386, 390, 392). It is difficult to describe results with data of statistical significance without considering the actual data treated. With these mean and standard deviation data, a statistically significant difference is unlikely. It is recommended to present tables and graphs with the data and statistics clearly exposed.

In Discussion.

The study fits more into another objective: This study aimed to evaluate awareness-raising interventions in two types of health centers, not to determine the effectiveness (line 394). References to similar results correspond to reviews, so it is difficult to compare according to a result presented in this study (line 398, 399). Also, change general favourable to generally favorable.

Regarding the rebound effect, the explanation is not sustainable with the results of the study and they are due to unpredictable general contexts that can be confirmed on this occasion (line 399 - 409). The limitations of the study are important and therefore it is necessary to have empirical data on it to be analyzed in light of what is presented in this manuscript. For example, can the data on attrition or non-completion of the process significantly affect the results? (line 426-437). The scopes indicated in lines 410-425, rather justify this study, for which they should be in the initial introductory context. Here the discussion must be in relation to the results obtained and how they contribute or contribute to reducing discrimination or stigmatization in the context of health centers.

The conclusions.

Here the objective changes, from ascertaining to generating consciousness (line 439). The procedures used assess the effect of the interventions in order to reduce stigmatization. We could not ensure the first and neither can reflection (440). In conclusion, the results should address whether the intention or claim of the authors is close to an objective as relevant as reducing stigmatization in the practice of health professionals.

Change favourable to favorable (line 443) and practise by practice

Author Response

Dear reviewer,

Thank you very much for your advice that we have used to improve our manuscript.

Please note that we use British English. Therefore, we cannot consider some of the spelling changes that you propose since they are from American English and not correct in British English. In addition, please also note that the second reviewer asked us to use tables instead of graphs to display the results.

Please see the attached point-by-point response table in which we detail one by one all the changes made in reference to each of your comments.

We look forward to your comments.

Sincerely,

The authors

Reviewer 2 Report

Dear authors,

Congratulations on the completion of the paper.
I think it is a paper with interesting content that can contribute to raising awareness of the need to reduce the stigma of people with mental health issues.

I think that the article is suitable for publication in its current state.

Sincerely.

Author Response

Thank you for your generous advice during the review process.